# Polymer-Based Additive Manufacturing: Process Optimisation for Low-Cost Industrial Robotics Manufacture

**DOI:** 10.3390/polym13162809

**Published:** 2021-08-21

**Authors:** Kartikeya Walia, Ahmed Khan, Philip Breedon

**Affiliations:** 1Department of Engineering, Nottingham Trent University, Nottingham NG11 8NS, UK; kartikeya.walia@ntu.ac.uk; 2PepsiCo Europe, Leicester LE4 1ET, UK; ahmed.khan@pepsico.com

**Keywords:** industrial robotics, low cost, additive manufacturing, polymer materials

## Abstract

The robotics design process can be complex with potentially multiple design iterations. The use of 3D printing is ideal for rapid prototyping and has conventionally been utilised in concept development and for exploring different design parameters that are ultimately used to meet an intended application or routine. During the initial stage of a robot development, exploiting 3D printing can provide design freedom, customisation and sustainability and ultimately lead to direct cost benefits. Traditionally, robot specifications are selected on the basis of being able to deliver a specific task. However, a robot that can be specified by design parameters linked to a distinctive task can be developed quickly, inexpensively, and with little overall risk utilising a 3D printing process. Numerous factors are inevitably important for the design of industrial robots using polymer-based additive manufacturing. However, with an extensive range of new polymer-based additive manufacturing techniques and materials, these could provide significant benefits for future robotics design and development.

## 1. Introduction

Robots are deployed for numerous applications and in various industries, with a continuous demand for more companies and manufacturers trying to integrate robotics and automation into their production lines [1,2,3]. This wide adoption has also seen a reduction in cost for industrial robots, but the entry barrier of cost [4] is still considered high for some applications. These applications can include the food, packaging, and electronics industries, where the payload can be relatively lightweight [5,6]. There are many potential applications where the industries outlined above can benefit if the appropriate robotic solutions are available [7].

When a robotic system is tasked to handle light payloads, these systems can often be over-specified based on the available ‘standard robot commercial’ specifications. When such a system is over-specified based on existing commercial availability, this can result in redundancy and an investment in a system that is over-engineered for the task specified. Optimisation of the material selection and a design specification based on a low payload can result in the design of a lighter robotic system [8,9].

With the development of innovative manufacturing solutions, additive manufacturing (AM), or more popularly known as 3D printing, provides a realistic approach to the design of lightweight and customised designs [10,11]. In addition, its throughput and low-cost approach for initial prototyping solutions have drawn increasing research interest during the last decade. Integrating polymer-based 3D printing for the purposes of manipulator fabrication with lightweight applications is therefore central for further investigation in relation to this study.

According to the ISO/ASTM 52900:2015 [12], additive manufacturing processes have been broadly classified into seven categories: (1) binder jetting; (2) directed energy deposition; (3) material extrusion; (4) material jetting; (5) powder bed fusion; (6) sheet lamination; and (7) vat photopolymerisation. The design specification and the choice of material influence the properties of the output [13,14]. A wide availability of materials, with different mechanical and thermal properties, for various AM processes provides better control of the desired characteristics of the design [15,16,17].

Geometric Dimensioning and Tolerancing (GD&T) is a protocol [18] at the center of any mechanical design and is complimented by a multitude of manufacturing processes and material selection allowing for the exploitation of numerous opportunities. A CAD model’s mesh file manufactured using any of the above-mentioned different 3D printing processes tends to differ in dimensions, usually in a range of less than ± 0.5 mm [19,20]. However, this value still represents a significant variation in dimensions for the robotics applications or for any other application requiring assembly of various 3D printed or off-the-shelf components. Hence, this dimensional deviation needs to be compensated [21,22,23] for at the design stage which adds to the overall design complexity. There has been some effort to standardise the GD&T characteristics of additively manufactured parts [24,25,26].

## 2. Materials and Methods

This section discusses in-depth the various polymer-based AM processes and several variables that should be considered while selecting a preferable manufacturing method.

### 2.1. From Digital Model/CAD to a 3D Printable Mesh

Mesh is a digital blueprint of the 3D CAD model which encompasses the geometric data for that part. Most of the 3D modelling software has an option to export a mesh file in the latest updates because of the growing acceptance of 3D printing for rapid prototyping in every manufacturing industry. Some used mesh formats [27] include .stl, .obj, .amf, and .3mf. All these file formats have gained respectable support across the 3D printing toolchain, but all of these vary in terms of the type of data they store and what information goes to the 3D printer. The choice of the file format is also tightly coupled with the tool being used for 3D printing.

#### 2.1.1. Standard Tessellation/Triangulation Language (STL)

The most commonly used [28] file format, standard tessellation/triangulation language (.stl), essentially divides a 3D model surface into smaller triangular meshed surfaces. The triangles can be made arbitrarily small to approximate the curved regions, but increasing mesh density increases the file size. In Figure 1, the perfect spherical surface on the left is approximated by tessellations. Figure 1c uses big triangles, resulting in a coarse model with a small file size, whereas Figure 1b uses smaller triangles and achieves a smoother approximation at the cost of a much larger file size [29].

The .stl format will soon become obsolete [30], as it is the most bloated file format for storing mesh data. It stores the normal vectors to the triangles even though this is redundant information. Additionally, there is no information about the inter-connectivity of the triangles, and therefore a watertight manifold of the mesh cannot be assured; it hence require mesh repairs. The .stl format also lacks the capability of storing units, material, and texture of the design.

#### 2.1.2. OBJ

The .obj format is the preferred mesh format for multicolour 3D printing applications as it can store colour and material for the part. Its open-source nature and the ease of use have made it the second most used format. However, as with .stl, there is a balance required between the file size and the mesh accuracy as it also depends on the polygon tessellation of the surface. The texture and colour of every 3D surface are mapped on a 2D contour and stored in a companion file called Material Template Library (.mtl) format. Advanced schemes exist to store curves or free form surfaces [31] without losing any precision, which makes it a little more complicated [32] to repair or debug if the file has errors.

#### 2.1.3. AMF

This format was introduced by the American Society of Testing and Materials (ASTM) as a replacement for the bloated and error-prone .stl format [33,34]. It being a .xml-based format, natively supports geometry, scale, colour, materials, lattices, duplicates, and orientation. Similar to the .stl format, .amf also stores tessellated triangles, but with better accuracy [35]. Additionally, it can also allow for curved triangles, which reduces the number of facets. As an encoding-based framework, it can allow for the repetition of similar geometry without bulking up file size [36]. However, this format has not been widely adopted in the 3D printing industry.

#### 2.1.4. 3MF

After being developed internally by Microsoft with some inspiration from .amf, the .3mf format was launched separately. Microsoft created a 3MF consortium [37], which includes all the big players in the 3D printing industry—Autodesk, Stratasys, Ultimaker, Materialize, Shapeways, 3D Systems, Dassault Systems, Siemens, HP, and GE to name a few. All these stakeholders govern the further development and progress of the 3MF format and in turn increase adoption of .3mf. The involvement of most of the key stakeholders in this field make it an inevitable format of choice to replace .stl format. 3MF is a complete and simple .xml-based format that is unambiguous and free for implementation [27]. It also overcomes the issue of a watertight mesh by making sure that the files have a 100% manifold, no overlapping triangles, and no cracks. A GitHub repository by Microsoft gives open access to everyone for contribution.

Choosing the correct format for this application in robotics is imperative as this affects the toolchain, production efficiency, and the quality of the prints. Additionally, different robotics’ applications require different tolerances for assembly, depending on the components and assembly required post-fabrication. For example, print-in-place (non-assembly) functional components [38] require higher details, specifically lower surface roughness and better curvature information compared to the non-functional part. The faceting of a curvature directly affects the friction between two mating parts in an assembly and hence, the performance at the joint. A cumulative effect of this can lead to lower overall performance in the assembly. Another example where the mesh information is of importance is when minute details such as mechanical threads or transmission devices such as gears [39] are to be fabricated, as dimensional inaccuracies can lead to backlash and consequently restrict the performance. Therefore, it is important to select the suitable mesh format which captures sufficient detail.

### 2.2. From Mesh to Print

Slicing is principle terminology in the domain of 3D printing as it is a tool that is specific step in the process but fundamentally important to the whole process of printing, immaterial of the hardware. For a mesh (Figure 2a) to be additively manufactured, the 3D model must be broken down into smaller definitions for the hardware to interpret and print. Since most of the printing processes 3D print a model layer by layer, dividing the mesh into these layers or slices (Figure 2b) is key [40].

These slices are then used to generate a toolpath, which is interpreted by a 3D printer to begin the manufacturing process.

A novel alternative to the slicing is the voxel (3D pixel) [41] generation in which instead of dividing the model into thin layers, it is broken down into small volumetric pixels (Figure 2c), which are then used to generate the toolpath.

Amongst many other parameters, the thickness of the layers and the size of the voxels also decide the quality of the final product. Homogeneity of the material in the printed parts becomes an essential factor for robotics and it can be observed that voxels provide the highest quality and control of material properties in all directions. Although slicing is well accepted in most applications, it is also successful as compensation is undertaken either by selecting better/stronger materials or by decreasing the layer resolution for improved fusion.

Several other 3D printing parameters which are key to the slicing process are discussed in detail in the next section.

### 2.3. 3D Printing Parameters

All the various 3D printing technologies have selected parameters that influence and define the quality of the printed object/part [42,43]. The tangible measures of the quality for a 3D print include surface finish, roughness, overall part strength, axis specific strength, thermal stability, watertightness, dimensional accuracy, etc. All these can features can be very closely controlled by manipulating the printing parameters.

There is a large list of parameters for the various technologies; however, there are particular parameters which are common and contribute to visible differences (Table 1).

#### 2.3.1. Layer Height

Layer height can be described as the thickness of each layer of the 3D print. It controls the coarseness of the part surface and affects the print resolution and the dimensional precision along the Z axis. It also significantly affects the print duration. An improvement in detailed parts can be produced with higher resolution or lower layer thicknesses. A lot of research is going on for optimising the layer thickness parameter and developing adaptive slicing techniques [44,45,46,47]. Different printers have different maximum resolutions, which vary from 70 microns in FDM to as low as 10 microns in MJF, SLA, and SLS technologies. This is one parameter that is adjustable in all the technologies. Several studies have been conducted to evaluate the impact and influence of layer thickness onto the mechanical properties of the structure [42,48,49,50,51,52,53,54,55].

#### 2.3.2. Infill

Infill defines the degree of filled volume inside the part and essentially decides the density of the part. Infill is mostly useful during the prototyping stages, when the part does not have a functional utility and economical and faster prints are required just for the visual purposes or geometric dimensional and tolerance analysis post-fabrication. Higher infill provides more structure and rigidity against deformation under impact. In spite of the fact that in most 3D printing technologies infill percentage is typically not an editable parameter, in FFF/FDM it can be experimented with a lot. Infill parameter has been one of the widely researched and experimented domains in additive manufacturing, with many publications [48,49,50,51,52,53,54,55,56,57,58,59,60,61,62,63], suggesting its direct impact on the mechanical properties of the output with different settings and parameters selected for different materials.

#### 2.3.3. Shell Thickness

Shell is the outermost surface of the part including the walls inside that support it for strength [51,52]. The thickness parameter for shell essentially comes into the picture and becomes crucial when infill % has been varied to save on material. This provides robustness to the part and makes it stronger against impacts. This setting is usally editable for the prints done using FFF/FDM technology.

#### 2.3.4. Supports

3D printing ‘overhung design’ features are a critical aspect to be taken into consideration. Usually, the next layer builds upon the previously printed layer, or in some cases it is encapsulated in the raw material itself. However, in some technologies such as FDM and SLA printing, overhangs are a limitation as there is nothing surrounding and supporting these portions. This is overcome by generating supports during the slicing stage. The supports are the removable structures that allow for overhangs to be printed without affecting the quality of the part. By definition, this requires extra printing material, additional print time, and further post-processing steps. To optimise support generation, some efficient algorithms [64] and generative design [65] can be used.

#### 2.3.5. Printing Orientation

This is a very crucial setting for the user, as orientation is a decisive parameter for the ultimate success of a viable print. Optimum printing orientation enables successful printing in a smaller time duration and ensures required strength of part(s) along a specific axis. Orientation is also modified according to the geometric importance of some features in the model. Printing orientation also impacts the mechanical strength of the printed part [48,50,54,60] along various axes of load application. This also greatly decides the amount of support generation for overhangs and as a result the amount of printing material to be used.

### 2.4. Various Polymer-Based AM Processes

The ASTM ISO classification of 3D printing process broadly classifies all the technologies. Within the last decade, there have been several sub-classifications of those categories as novel ways of additive manufacturing are invented and older processes have evolved [66]. The discussion has been kept limited to polymers and polymer composite technologies (Figure 3) in light of the use case application being focused on, i.e., lighter and affordable robotics.

#### 2.4.1. Vat Polymerisation

Vat polymerisation selectively cures photopolymer resin in a vat using a light source. Three different novel processes (Figure 4) that use this technology have been developed.

Stereolithography (SLA). This was the very first 3D printing technology to be developed and was invented by Chuck Hull in 1986 [68]. It was later patented and commercialised. Technologically, it uses a pair of galvanometers with mirrored surfaces to deflect an incident laser (mostly solid-state ultra-violet) and trace a 2D cross-section on the transparent vat filled with photosensitive resin. The solidified part keeps rising gradually in the Z axis after completion of each layer until the whole part is printed. The resolution in the Z axis can be as low as 25 microns, which creates very smooth surface finishes. A very high degree of resolution and complexity is achievable in the X-Y axes because of a very minute laser spot size, for instance, 85 microns in Formlabs [69] machines. However, a point laser takes longer to trace the cross-section of an object compared to DLP.

Digital Light Processing (DLP). As its name suggests, this technology uses a digital light source, a conventional lamp, to cure the parts, instead of using laser UV. The rest of the process is similar to SLA but is comparatively faster [70]. Since the projector is a digital screen, the image of each layer is composed of square pixels, resulting in a layer formed of small rectangular blocks all at once. DLP produces highly accurate parts with excellent resolution similar to that of SLA. It requires a shallow vat of resin compared to SLA, hence resulting in reduced waste and lower running costs.

Continuous-DLP (CDLP). In SLA and DLP processes, there is a pause interval after every layer being printed. This is required to allow for more resin to flow into the projectable area and to continue printing the next layer. This makes the whole process discontinuous and consequently slow. Continuous-DLP, also known as Continuous Liquid Interface Production (CLIP) and Carbon Digital Light Synthesis (CDLS), subdues this issue [71,72] by utilising a chemical property of the resin and making appropriate changes in the printing process. It uses digital light projection in combination with oxygen-permeable optics and programmable liquid resin [73]. A second heat-activated programmable chemistry in the material enables the production of high-resolution parts with isotropic and high-grade mechanical properties. A sequence of UV images is projected onto the UV-curable resin through an oxygen-permeable window, which solidifies the part, and the platform then rises. A dead zone is enabled, which is a thin, liquid interface of uncured resin between the window and the printing part. This allows for more resin to flow under the solidified part in the dead zone and be available for the next layer, making the whole process continuous.

#### 2.4.2. Material Extrusion


Fused Deposition Modelling (FDM)/Fused Filament Fabrication (FFF)


Printers using this technology (Figure 5) are the most widespread and popular [74,75] in the market. FDM is an extrusion-based process, where a solid filament such as plastic (PLA, ABS, etc.) is pulled into an extruder and melted by a heated nozzle and deposited into the required form to draw a thin cross-section layer. In the gantry-type Cartesian 3D printers, the extruder head has a movement restricted to two axes, usually planar X and Y axes, and the vertical movement is in the Z axis. Many printers have the bed movement in the Y axis with the extruder capable of traversing both horizontally and/ or vertically. This together enables precise numerically controlled movement and hence the development of the 3D prototype. This technology has a relatively smaller infrastructure and material cost.

#### 2.4.3. Material Jetting (MJ)

In the polymer (plastic)-based material jetting technology (Figure 6), also known as PolyJet, a photopolymer liquid is jetted on a build platform, using either a continuous or Drop on Demand (DOD) approach, which is immediately followed by exposure to a UV light source, which cures or hardens the form. After every layer, the build platform descends before the next pass. This is a swift and efficient method of producing solid cross-sections [24]. Since the print head is an array of many minute holes, similar to an inkjet printer, it opens the possibilities of multi-material and multi-colour 3D printing as well [25]. A high level of droplet control and positioning is possible and the droplets that are not used are recycled back into the printing system.

#### 2.4.4. Powder Bed Fusion

This technology uses raw material in the powdered form to build 3D objects by selectively fusing cross-sections of the 3D model bonding agents or lasers. The final parts printed using Powder Bed Fusion technology exhibit more consistent mechanical properties, fine feature resolution, and higher quality surface finishes. There are two developed processes for polymers (Figure 7), namely Multi Jet Fusion and Selective Laser Sintering.


Multi Jet Fusion (MJF)


MJF is an industrial 3D printing process used to rapidly produce functional prototypes and end-use production parts with nylon. It uses an inkjet array to selectively apply fusion/bonding and detailing agents across a bed of nylon powder in one pass, which are then fused by thermal treatment, using some heating elements, into a solid layer. This is followed by a re-coater applying a thin layer of the fresh powder onto the previously formed layer, for the process to continue. Since the part is enclosed in a chamber of unfused powder, there is no requirement to generate supports, therefore reducing the post-processing times. Although, the encapsulated part must be separated manually from the loose and residual powder by a process such as vacuuming or sandblasting.


Selective Laser Sintering (SLS)


Laser Sintering is a technique that enables the fabrication of an object by almost melting successive layers of powder together. It uses powder bed fusion technology and polymer powder, usually nylon, where subsequent cross-sections in a very thin layer are made by fusing together the powdered material [26]. The surrounding powder on the bed that has not been sintered acts as a temporary support until the full object is manufactured, hence eliminating the requirement of the extra support structure. The build time may depend upon the resolution of layers. The shortest axial dimension of the part to be printed should be oriented in the Z direction to minimise the total number of layers required. The cost and practical considerations are comparatively higher, but the print quality achieved is also superior to other technologies. The post-processing is similar to the MJF process.

Selecting the appropriate fabrication method is a decisive step in the whole design process. Specifically, for low-cost robotics, this becomes critical. Since lowering the cost of production is one of the aims, the overall quality of the produced parts and components is still paramount for industrial application. Every process mentioned in this section has, on average, a different costing and lead time for production, which adds to the number of variables to be considered.

### 2.5. Design Considerations for Various AM Processes

A good understanding of the design workflow for manufacturing processes and acknowledging the manufacturing constraints during the design stage are pivotal to the successful fabrication of the part. Meeting the design criteria, dimensions, and tolerances to create an efficient design is the first step in this process, also known as the concept design. Finite Element Analysis (FEA) has become an important tool for the optimisation step in the design process and ensures a certain factor of safety for functional designs of end use parts even before production.

An iterative design strategy is well accepted and has been in use since the very beginning of digital modelling. With the introduction of additive manufacturing, the physical production of the part has also been included in the design process. 3D printing is now being used to build functional parts but is also being used as part of the iterative design stages for validation. Numerous publications are available in relation to several strategies [76,77,78,79,80,81] for Design for Additive Manufacturing (DfAM).

Achieving a very high degree of complexity and detail in a final product has become a possibility that has led to the restructuring and diversification of design ideas. With the additional freedom in the design workspace, there are supplementary design considerations/limitations as well for additively manufactured forms.

#### 2.5.1. Print Orientation

Whilst not exactly a limitation, print orientation should be addressed at the design stage. As most of the printing processes produce parts layer by layer, the risk of failure via crack propagation is maximum in the direction orthogonal to the building process or along a layer. This can either be considered before slicing the mesh by orienting the model in such a manner to have layers along the direction of the least expected load or making necessary design changes such as chamfers and fillets for improved stress distribution.

#### 2.5.2. Feature Specific Considerations

Figure 8 shows various features that are important for AM design purposes. Recommended values for these features in design are given in Table 2.
Supported Walls: The walls that are connected to the rest of the print on at least two sides.Unsupported Walls: These are connected to the rest of the print on one side.Support and Overhangs: The maximum inclination a structure can be printed without support.Embossed and Engraved Details: Features on the model that are raised or recessed.Horizontal Bridge: The printable horizontal span without support.Holes: Minimum diameter of a printable hole.Connecting/moving Parts: Recommended clearance between two moving or connecting parts.Escape Holes: The minimum diameter for escape holes to allow for removal of build material.Pin Diameter: Minimum diameter a pin can be 3D printed at.Tolerance: Expected dimensional accuracy.Minimum Feature: The recommended minimum size of any feature to ensure it will not fail to print.

#### 2.5.3. Avoiding Supports

Supports are an auxiliary structure in a 3D print that helps to print overhangs. However, the generation of supports leads to increased print times and further material consumption. It is also a factor to be considered in relation to AM becoming a zero-waste-producing technology. Printing orientation is one parameter, as mentioned in Section 2.5.1, which not only affects the overall mechanical properties but also largely influences the number of supports required for the print [82,83]. Smarter designs inclusive of some specific features can also be used in some if not in all cases to avoid/reduce the number of supports in a 3D print.

For printing cantilever structures in a design, a chamfer of 30° to 45° with the vertical plane at the under edge will eliminate the overhang and hence the requirement of support.

The bridgeable roof structures fail to print without support if they have a hole feature (Figure 9b), making the bridge discontinuous. To overcome this, the hole can be closed from the bottom (Figure 9a) with just a minimal thickness of approximately 1 layer (100–300 microns). This will allow the overhang to be bridged appropriately, and the thin structure responsible for closing the hole can easily be removed later during post-processing.

Novel hardware capabilities are also being applied such as multi-axis 3D printing [84,85,86] and reusable supports’ [87] innovative part production methods such as object partitioning [88] and support slimming [89].

### 2.6. Material Libraries

The materials that are available for 3D printing are as diverse as the 3D printing technologies and the outcomes that result from the process [90]. Clearly, the freedom of design allows the manufacturers to determine the shape, texture, and strength of a product. However, these parameters can also be varied by choosing the correct material for a particular application scenario.

There are numerous ways to classify the various materials available, including their physical state: plastics, powders, metals, and resins. These are classified here according to the four AM processes that have been broadly discussed.

#### 2.6.1. Vat Polymerisation

This technology uses light-reactive thermoset materials called “resin.” Although the materials are limited to photosensitive thermosets, some efforts to mix particulate into the resin prior to processing have been successful [91,92,93]. When exposed to certain wavelengths of light, short molecular chains join together, polymerising monomers and oligomers into solidified rigid or flexible geometries. A wide range of photopolymer resin formulations have been created unique optical, mechanical, and thermal properties to match those of standard, engineering, and industrial thermoplastics [16].

The names of the resins are manufacturer-specific [94]; however, on the whole, there are materials that have different shore hardness values and are flexible, heavily filled with secondary materials like glass and ceramic, or permeated with mechanical properties such as high heat deflection temperature or impact resistance. Material ranges from industry-specific, such as material for dentures with biocompatibility, to those that closely match final materials for prototyping, formulated to withstand extensive testing and perform under stress.

#### 2.6.2. Material Extrusion

Amorphous and semicrystalline polymers are the main subcategories for the type of polymers used for material extrusion. The range and variety of materials available for material extrusion is too vast to be covered here in detail, but they are comprehensively mentioned.

The most commonly used materials for FDM are:PLA: Polylactic Acid; a preferred material due to ease of use and low cost;ABS: Acrylonitrile Butadiene Styrene; low-cost material for tough and durable parts;Flexible: TPU or TPE (Thermoplastic polyurethane and Thermoplastic elastomer respectively); known for elasticity;PETG: Polyethylene terephthalate glycol; ease of printability, smooth surface finish, and water resistance;Nylon: tough and semi-flexible material that offers high impact and abrasion resistance;PC: Polycarbonate; known for its strength, durability and a very high heat and impact resistance;PP: Polypropylene; high-cycle, low strength applications due to its fatigue resistance, semi-flexible, and lightweight characteristics;Composites: Metal filled, glow in the dark, magnetic, conductive, colour changing, biodegradable, clay, wax, etc. There are some composite materials available with carbon fibre and glass fibre micro-particles infused.

Apart from these common materials, there are several soluble materials that are used for printing supports:PVA: Polyvinyl alcohol; dissolves in water and is often used as a support material for complex prints;HIPS: High Impact Polystyrene; dissolves in d-Limonene, most used as a dissolvable support structure for ABS models.

A manufacturer known as Markforged [95] has developed a technology called ‘Continuous Fibre’ for inlaying continuous strands of materials like Carbon Fibre, Kevlar, and Fibreglass at mechanically critical portions in a print along with the usual filament of choice. This process significantly improves the mechanical properties of strength and stiffness [96], and the parts produced are suitable for industrial applications.

In addition, there is a wide range of industrial-grade reinforced filaments available with very high strengths, thermal resistances, and heat deflection temperatures, including Poly-ether-ether-ketone (PEEK) and Poly-ether Imide (PEI), etc.

#### 2.6.3. Material Jetting

Material Jetting uses proprietary photo-curable plastics and composites and a combination of these called “digital materials” [97]. Typically, these are photosensitive thermoset polymers that have a possibility for particle additions prior to deposition limited by the permissible viscosity for this technology [98]. Further research is currently being undertaken into expanding the range of materials that can be used with material jetting. Metals, ceramics, and silicones have already started to enter the market.

#### 2.6.4. Powder Bed Fusion

Limited to semicrystalline materials this process mainly utilises are nylon (PA: Polyamide) and TPU powdered materials for rigid and flexible solutions respectively. A few amorphous polymers such as polystyrene are also usable. Recently, a manufacturer called Sinterit has developed a compatible PP-based powdered material as well.

The nylon-based variants available are PA12, PA11, and PP. The variants available for flexible materials are TPU and TPE, with some combinations for different shore harnesses. Some PA-based composite materials are also available in the market.

## 3. Results

During the early design synthesis stage, there are many factors to be considered and decisions to be made based on the variables present in the AM process for a specific application. Many AM parameters are based purely on design decisions and are left to the manufacturer’s/designer’s potential. There are some parameters, including the choice of AM process and material, which can be qualitatively compared with cost vs. performance (quality) as the basis. A ‘Suitability Matrix’ is presented in Table 3.

Material selection is an important factor once the AM Process is finalised. The following charts (Figure 10 and Figure 11) show the qualitative comparison of several available materials for vat polymerisation and material extrusion technologies with cost v/s performance (durability and strength) as the basis. Discrete Weight Function (Equation (1)) is used for evaluating corresponding suitability values. A weight of 65% is given to the performance values and 35% to cost while calculating suitability. (For instance, a Suitability index of 10 will be given to a material with unmatched/ideal performance (10) available at no cost (0)).
(1)S=(0.65∗p)+(0.35∗(10−c))
S: Suitability Value; *p*: Performance Value; *c*: Cost Value.

This method of qualitative analysis illustratively helps in determining the best-fitting material for the given requirements of performance and cost.

## 4. Discussion

### 4.1. Significance of AM Integration with Robotics

With a process that is no longer bound by the limitations of conventional manufacturing methods, the introduction of industrial 3D printing technologies has brought a positive disruption in transforming design and manufacturing methods. Additive manufacturing relies on steady, repetitive motion to build each infinitesimal layer. Robotics is renowned for its repeatability and control. Both technologies complement each other, and this can be used to achieve positive results that can be shared with customers to provide an improved and more effective robotic system.

In 2018, consultancy McKinsey and Company published [99] “The Next Horizon for Industrial manufacturing: Adopting Digital Technologies in Making and Delivering”, which summarised the importance of disruptive technologies and their impact on industrial manufacturing. An excerpt from the report is mentioned below:

In the past few years, advanced industrial companies have made solid progress in improving productivity along the manufacturing value chain. In the U.S., for instance, the productivity of industrial workers has increased by 47% over the past 20 years. However, the traditional levers that have driven these gains, such as lean operations, Six Sigma, and total quality management, are starting to run out of steam, and the incremental benefits they deliver are declining. As a result, leading companies are now looking to disruptive technologies for their next horizon of performance improvement.

Two of those disruptions named in the McKinsey report were additive manufacturing (3D printing) and robotics. The intersection of both presents a newfound potential for robotics companies and their customers.

As the robotics design process can be complex with numerous design iterations, 3D printing is ideal for rapid prototyping. This begins with a concept and different design parameters that are used to meet an intended application or routine. Prototypes may be developed quickly, inexpensively, and with little overall risk, through a 3D printing process.

During the initial stage of robot development, exploiting 3D printing can not only improve the technological solution, but provide design freedom, customisation, improved responsiveness, and sustainability and provide overall cost benefits. With the availability of engineering grade and composite 3D printing materials, the ability to create custom parts uniquely suited for an industrial task is an open domain to be explored. Additionally, part consolidation [100] helps with easier and less time-consuming assemblies. Another huge benefit of using 3D printing for robots and robotic parts is producing smaller volumes with no upfront costs. Low-volume builds can be profitable as molds are not required and, in addition to the multitude of applications 3D printing, can support in relation to the robotics industry, provides wide-ranging potential for future innovation.

### 4.2. Adoption in Research and Industry

There are numerous 3D printable robotic arms [101], including BCN3D Moveo [102], Zortrax Robotic Arm [103], and HydraX [104], but most of these are imprecise and lack the robust design features required for industrial use.

One notable 3D printed robotics arm which has potential in industrial applications is ‘Dexter’. It was developed by Haddington Dynamics [105], and they initially prototyped the whole robot using PLA parts which were later replaced by 3D printed parts made with micro carbon-fibre-infused nylon material. As individual components were stronger, component volume was reduced in comparison with cutting individual parts saving on materials and therefore overall costs. Additionally, they were able to consolidate parts together and bring down the number of individual parts required, hence reducing assembly time. Another interesting research domain for the application of polymer-based additive manufacturing is in soft robotics [17,106,107,108,109], which is finding industrial suitability for end effectors.

## 5. Conclusions

With an extensive range of new polymer-based additive manufacturing solutions, there are many potential benefits linked directly to the future of robotics design and development.

Numerous considerations are inevitably important when considering the design of industrial robots using polymer-based additive manufacturing, but as a process, it clearly shows promise in delivering realistic, low-cost robotics solutions. This can be approached by weighting several factors to derive a suitability index that can be used to compare various features/specifications of interest (comparing qualitative data quantitatively) and making an informed decision for the suitable manufacturing method. Ultimately beneficiaries could include the food and plastic products industries, as currently a hesitation for investment in current industrial robotic solutions is clearly evident (Figure 12).

## Figures and Tables

**Figure 1 polymers-13-02809-f001:**
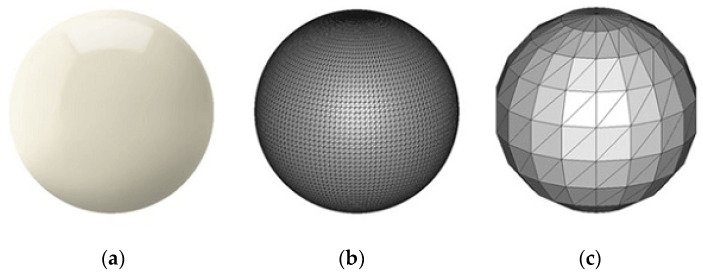
(**a**) CAD, (**b**) high-resolution mesh, (**c**) low-resolution mesh.

**Figure 2 polymers-13-02809-f002:**
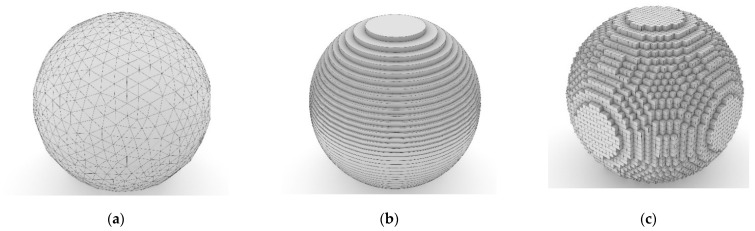
(**a**) Triangulated mesh, (**b**) 3D-sliced, (**c**) 3D voxels.

**Figure 3 polymers-13-02809-f003:**
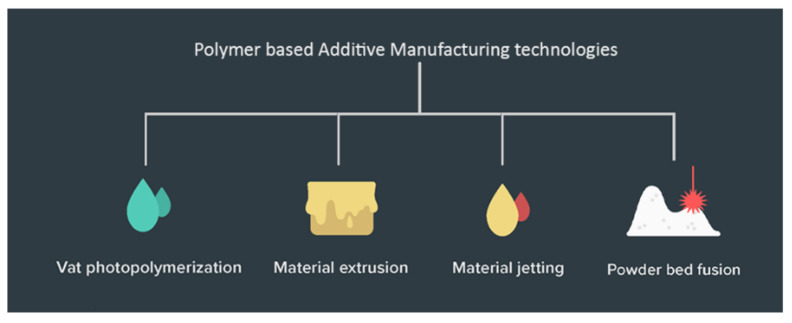
Types of polymer-based AM Technologies.

**Figure 4 polymers-13-02809-f004:**
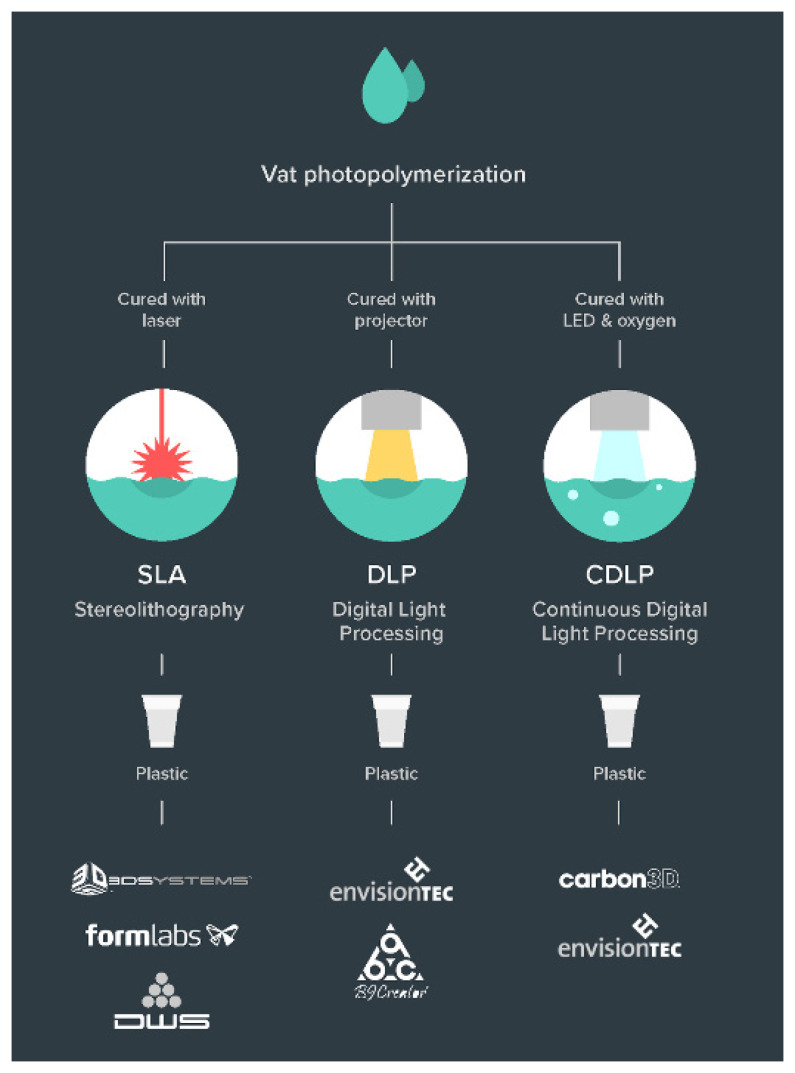
Types of vat photopolymerisation technologies [67].

**Figure 5 polymers-13-02809-f005:**
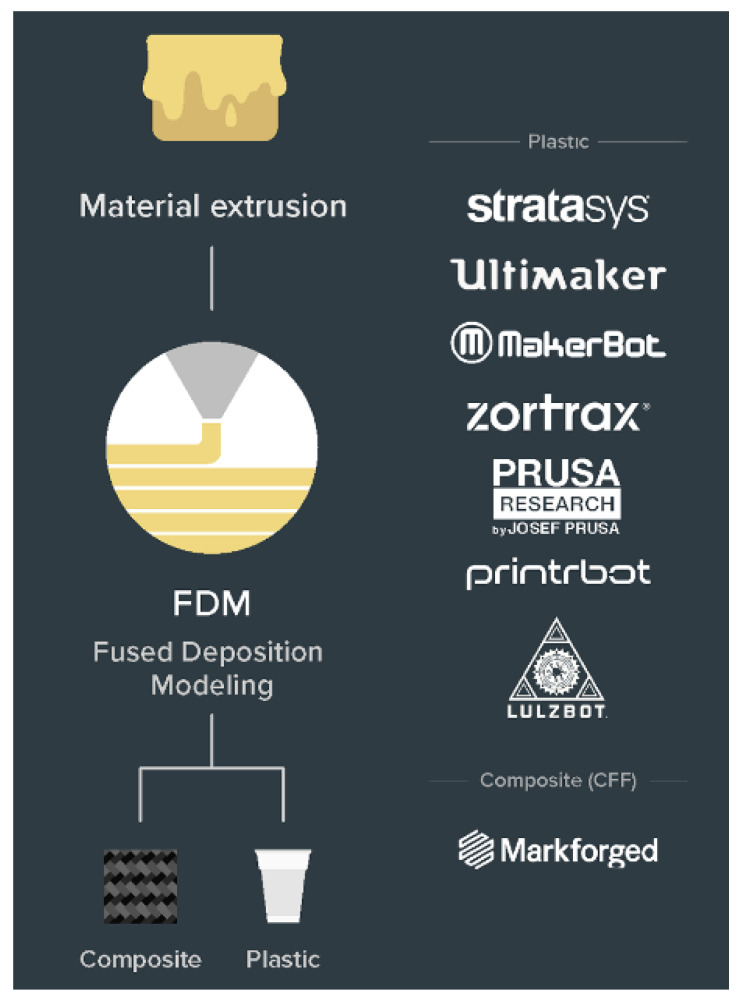
Material extrusion technology [67].

**Figure 6 polymers-13-02809-f006:**
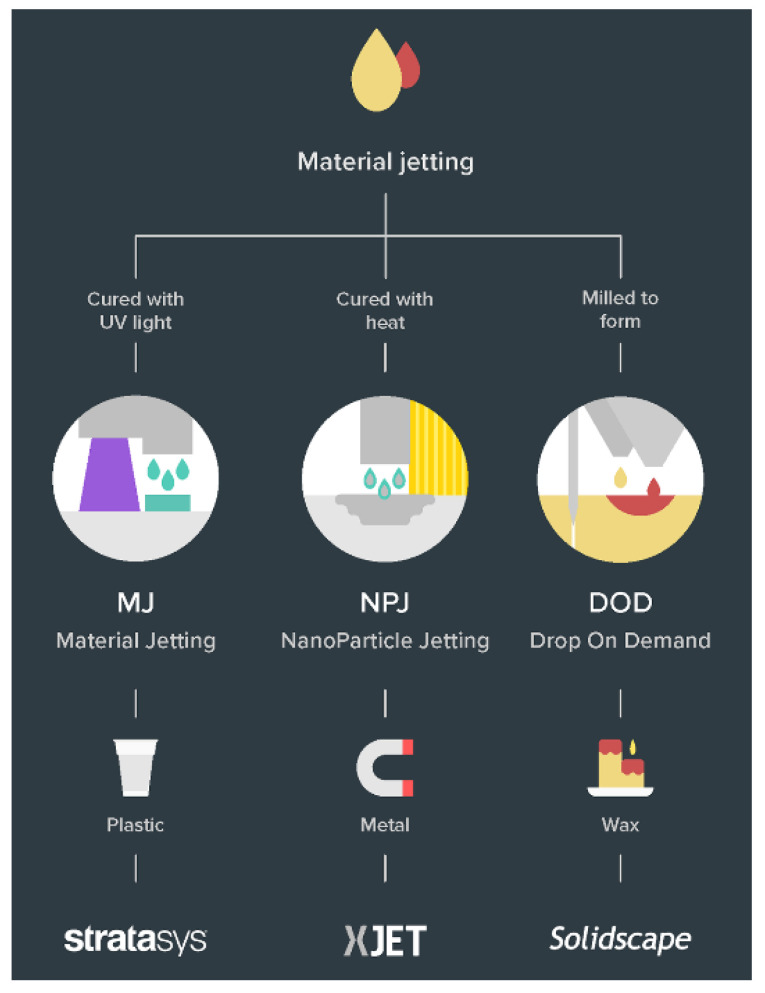
Material jetting technologies [67].

**Figure 7 polymers-13-02809-f007:**
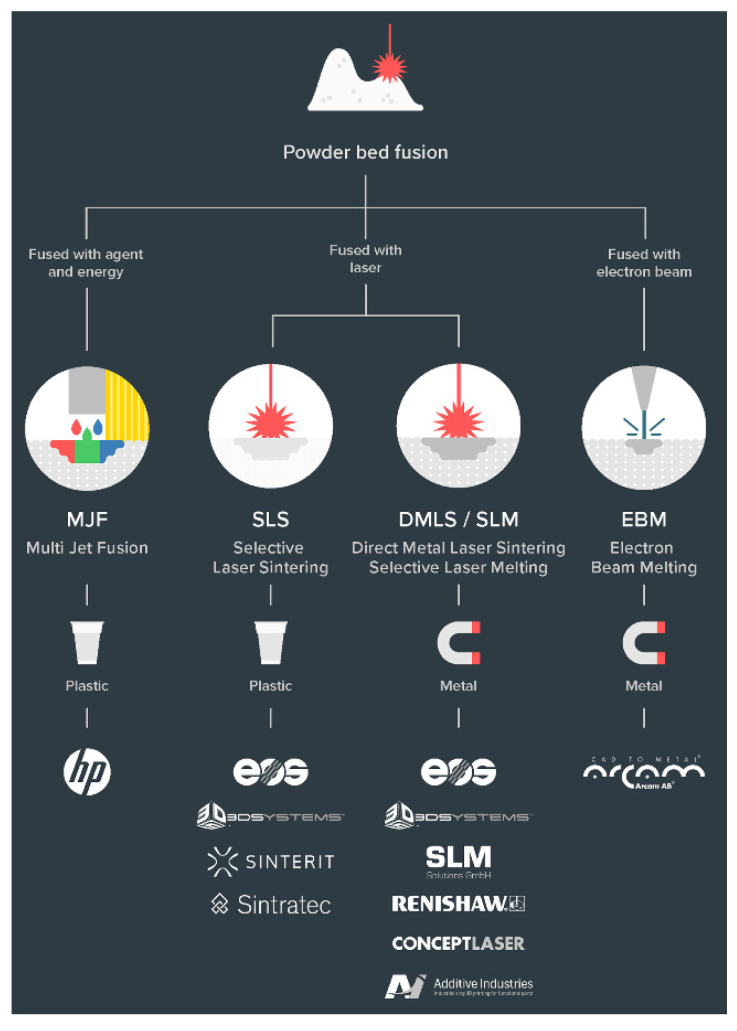
Powder bed fusion technologies [67].

**Figure 8 polymers-13-02809-f008:**
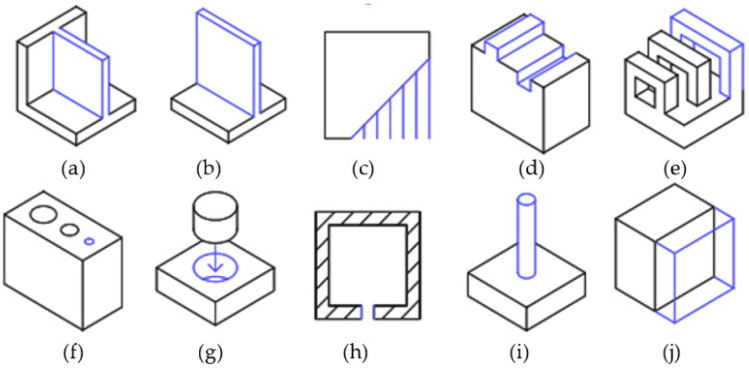
(**a**) Supported walls, (**b**) unsupported walls, (**c**) support and overhangs, (**d**) embossed and engraved details, (**e**) horizontal bridges, (**f**) holes, (**g**) connecting/moving parts, (**h**) escape holes, (**i**) pin diameters, (**j**) tolerance.

**Figure 9 polymers-13-02809-f009:**
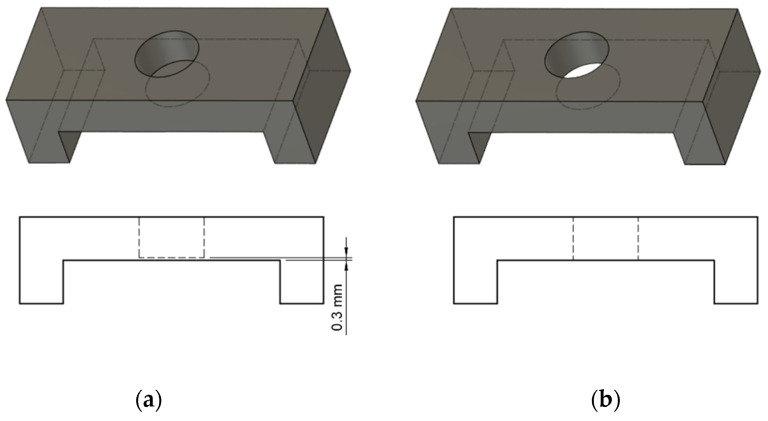
(**a**) Bridge feature with a closed hole, (**b**) bridge feature with hole.

**Figure 10 polymers-13-02809-f010:**
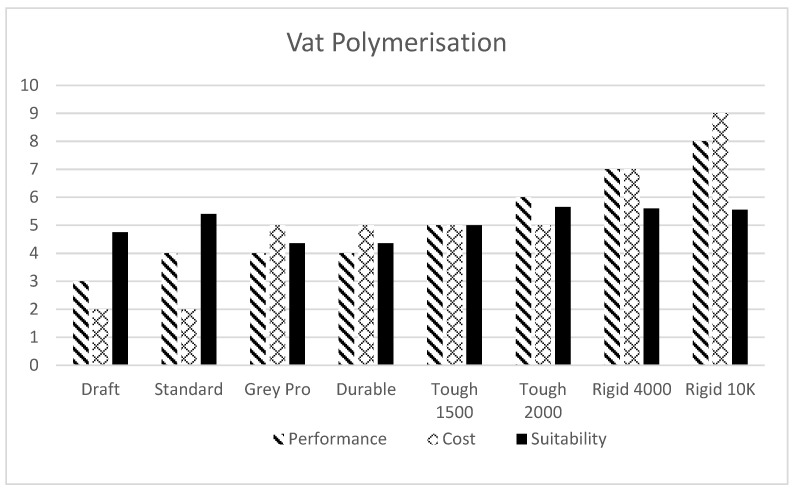
Comparison of various materials [94] available for vat polymerisation AM process.

**Figure 11 polymers-13-02809-f011:**
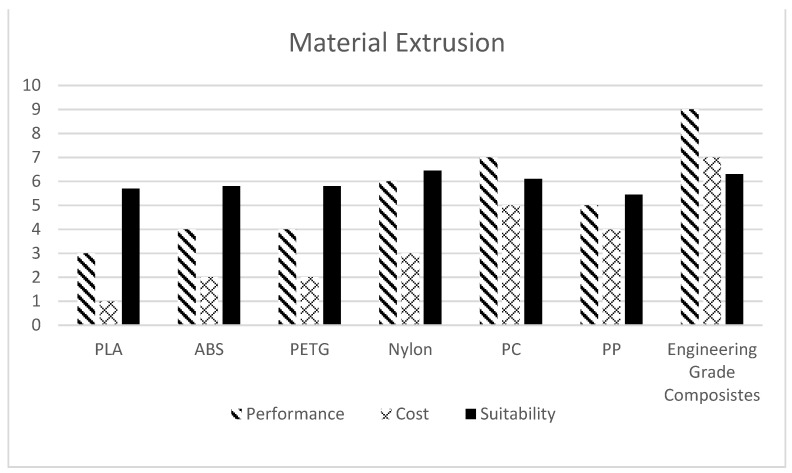
Comparison of various materials available for Material Extrusion AM Process.

**Figure 12 polymers-13-02809-f012:**
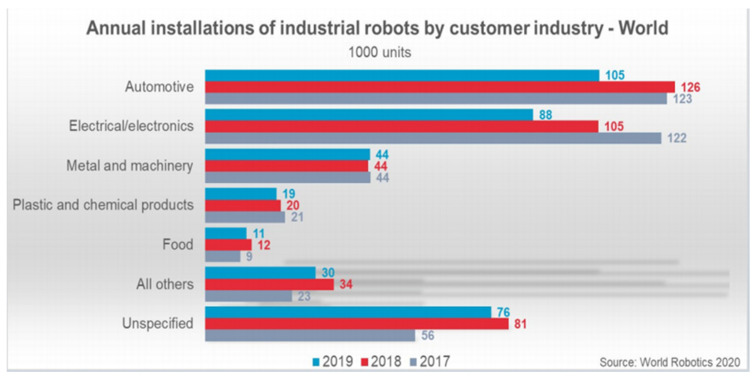
Adoption of robotics in various industries [110].

**Table 1 polymers-13-02809-t001:** Common 3D printing slicing parameters and effects.

Parameter	Effect
Layer Height	Print Resolution Dimensional Accuracy along Build DirectionPrint Duration
Infill	Weight (Material Consumed) Mechanical Strength and Rigidity
Shell Thickness	Mechanical Toughness Surface Hardness
Supports	Printability Print Duration Material Consumed Surface Finish
Printing Orientation	Printability Required Supports

**Table 2 polymers-13-02809-t002:** Recommended feature-specific values for several polymer-based 3D printing technologies.

Feature	Vat Polymerisation	Material Extrusion	Material Jetting	Powder Bed Fusion
Supported Walls	0.4 mm	0.8 mm	1 mm	0.7 mm
Unsupported Walls	0.4 mm	0.8 mm	1 mm	0.8 mm
Supports & Overhangs	Support always required	45°	Support always required	No support required
Embossed & Engraved	0.4 mm wide	0.6 mm wide	0.5 mm wide	0.5 mm wide
	0.4 mm high	1 mm high	0.5 mm high	0.6 mm high
Horizontal Bridge	21 mm	10 mm	0 mm	Not applicable
Hole	0.8 mm	2 mm	0.5 mm	1.5 mm
Connecting/Moving Parts	moving: 0.4 mm	moving: 0.5 mm	moving: 0.2 mm	moving: 0.2 mm
	connecting: 0.2 mm	connecting: 0.3 mm	connecting: 0.1 mm	connecting: 0.1 mm
Escape Holes	3.5 mm	Not applicable	Not applicable	5 mm
Pin Diameter	0.5 mm	3 mm	0.5 mm	0.8 mm
Tolerance	±0.5% (lower limit ± 0.15 mm)	±0.5% (lower limit ± 0.5 mm)	±0.1 mm	±0.3% (lower limit ± 0.3 mm)
Minimum Feature	0.2 mm	2 mm	0.5 mm	0.8 mm

**Table 3 polymers-13-02809-t003:** Suitability Matrix for various AM processes for low-cost robotics application.

AM Processes	Cost	Performance	Suitability
Vat Polymerisation	****	****	****
Material Extrusion	**	**	***
Material Jetting	****	*****	*****
Powder Bed Fusion	*****	****	***

An asterisk (*) is used as a marker to highlight the potency of suitability for that fea-ture to robotics applications (Five * being the best).

## Data Availability

Not applicable.

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
