# Peer review of "Polymer-Based Additive Manufacturing: Process Optimisation for Low-Cost Industrial Robotics Manufacture"

_polymers, 2021, doi:10.3390/polym13162809_

Round 1
Reviewer 1 Report
- Interesting paper.
- The study carried out shows the importance to consider different parameters/factors for the design of industrial robots using polymer-based Additive Manufacturing.
- The paper presents an approach of weighting several factors to derive a suitability index that can be used to compare various features/specifications of interest (Comparing qualitative data quantitatively) and making an informed decision for the suitable manufacturing method.
The paper has a very well-organized structure and a very logical sequence.
Author Response
The authors would like to express their sincere thanks to the Reviewer for their insightful comments and suggestions.
Reviewer 2 Report
First of all, it is a good manuscript to share the knowledge about factors to be considered upon the use of polymer 3D printers.
Hence, I personally think that the title of the manuscript is not quite appropriate. This is because the manuscript is more like to be printer weighting selection and printing limitation. Therefore I would suggest the author revise the title.
A table of slicing parameters is highly recommended, short paragraph is not attracting.
All the printer figures may put together for easier identification and comparison.
Similar to the 2.5.2's tables, compile in one table to look more tidy
Author Response
The authors would like to express their sincere thanks to the Reviewer for their in-depth comments, suggestions and corrections. The following changes have been made in accordance with the suggestions given by the reviewer:
- "First of all, it is a good manuscript to share the knowledge about factors to be considered upon the use of polymer 3D printers.Hence, I personally think that the title of the manuscript is not quite appropriate. This is because the manuscript is more like to be printer weighting selection and printing limitation. Therefore, I would suggest the author revise the title."
The title has been appropriately changed as per suggested by the reviewer. We hope that the title now accurately reflects the significance of polymer based additive manufacturing as a process that should be considered for robotics manufacture.
- "A table of slicing parameters is highly recommended, short paragraph is not attracting."
‘Table 1.’ Has been added to the revised manuscript highlighting the effect of the slicing parameters.
- "All the printer figures may put together for easier identification and comparison."
This was the initial thought for the original manuscript but as the combined image was illegible because of scale limitations, it was split into individual figures. ‘Figure 3.’ has been added to the revised manuscript to illustratively indicate the polymer based additive manufacturing technologies being discussed in the paper.
- "Similar to the 2.5.2's tables, compile in one table to look more tidy"
‘Table 2.’ has been updated and combined to include all the tables for the recommended values for several design features.
We hope we were able to acknowledge the comments satisfactorily.
Thanking you.